# From Forensics to Clinical Research: Expanding the Variant Calling Pipeline for the Precision ID mtDNA Whole Genome Panel

**DOI:** 10.3390/ijms222112031

**Published:** 2021-11-06

**Authors:** Filipe Cortes-Figueiredo, Filipa S. Carvalho, Ana Catarina Fonseca, Friedemann Paul, José M. Ferro, Sebastian Schönherr, Hansi Weissensteiner, Vanessa A. Morais

**Affiliations:** 1VMorais Lab—Mitochondria Biology & Neurodegeneration, Instituto de Medicina Molecular João Lobo Antunes, Faculdade de Medicina, Universidade de Lisboa, 1649-028 Lisbon, Portugal; filipe.figueiredo@medicina.ulisboa.pt (F.C.-F.); filcarvalho.uc@gmail.com (F.S.C.); 2NeuroCure Clinical Research Center, Charité—Universitätsmedizin Berlin, 10117 Berlin, Germany; friedemann.paul@charite.de; 3José Ferro Lab—Clinical Research in Non-communicable Neurological Diseases, Instituto de Medicina Molecular João Lobo Antunes, Faculdade de Medicina, Universidade de Lisboa, 1649-028 Lisbon, Portugal; acfonseca@medicina.ulisboa.pt (A.C.F.); jmferro@medicina.ulisboa.pt (J.M.F.); 4Serviço de Neurologia, Hospital de Santa Maria, Centro Hospitalar Universitário Lisboa Norte, 1649-035 Lisbon, Portugal; 5Experimental and Clinical Research Center, Charité—Universitätsmedizin Berlin and Max Delbrück Center for Molecular Medicine, 13125 Berlin, Germany; 6Institute of Genetic Epidemiology, Department of Genetics and Pharmacology, Medical University of Innsbruck, 6020 Innsbruck, Austria; sebastian.schoenherr@i-med.ac.at

**Keywords:** mitochondrial DNA, next-generation sequencing, massively parallel sequencing, whole genome sequencing, Precision ID, Thermo Fisher Scientific, variant calling, mixture, performance metrics

## Abstract

Despite a multitude of methods for the sample preparation, sequencing, and data analysis of mitochondrial DNA (mtDNA), the demand for innovation remains, particularly in comparison with nuclear DNA (nDNA) research. The Applied Biosystems™ Precision ID mtDNA Whole Genome Panel (Thermo Fisher Scientific, USA) is an innovative library preparation kit suitable for degraded samples and low DNA input. However, its bioinformatic processing occurs in the enterprise Ion Torrent Suite™ Software (TSS), yielding BAM files aligned to an unorthodox version of the revised Cambridge Reference Sequence (rCRS), with a heteroplasmy threshold level of 10%. Here, we present an alternative customizable pipeline, the PrecisionCallerPipeline (PCP), for processing samples with the correct rCRS output after Ion Torrent sequencing with the Precision ID library kit. Using 18 samples (3 original samples and 15 mixtures) derived from the 1000 Genomes Project, we achieved overall improved performance metrics in comparison with the proprietary TSS, with optimal performance at a 2.5% heteroplasmy threshold. We further validated our findings with 50 samples from an ongoing independent cohort of stroke patients, with PCP finding 98.31% of TSS’s variants (TSS found 57.92% of PCP’s variants), with a significant correlation between the variant levels of variants found with both pipelines.

## 1. Introduction

Mitochondria are the primary source of cellular ATP, while also prominently contributing to cell survival, differentiation, and apoptosis. They contain their own double-stranded circular DNA (mtDNA), with 16,569 base pairs (bps) in humans, responsible for encoding 22 transfer RNAs, two ribosomal RNAs, and 13 essential proteins of the oxidative phosphorylation (OXPHOS) chain. In comparison with nuclear DNA (nDNA), mtDNA has a higher copy number per cell (polyploidy), replicating independently of the nuclear genome, which allows for the existence of multiple genotypes within the same cell (heteroplasmy). Additionally, mtDNA shows a higher mutation rate due to the absence of protective histones, exposure to reactive oxygen and nitrogen species, and more rudimentary repair systems [1,2].

The peculiar nature of mtDNA has hindered the development of specific data analysis guidelines, since the large majority of datasets, guidelines, and bioinformatic pipelines for variant discovery analysis in next-generation sequencing (NGS)/massively parallel sequencing (MPS) are primarily, and sometimes exclusively, focused on nDNA [3,4], with mtDNA NGS analysis lagging years behind [5]. Out of the existing bioinformatic tools for mtDNA, with quality control assessment, variant calling, and haplogroup assignment [6,7,8,9,10,11,12,13,14,15,16,17,18,19,20,21], very few incorporate user accessibility, integrative data analysis, and regular updates. A suitable option is mtDNA-Server [17], which allows for FASTQ and BAM files as input and, considering the revised Cambridge Reference Sequence (rCRS) [22] as a reference, identifies heteroplasmic and homoplasmic variants, assigns a haplogroup with HaploGrep 2 [18], based on PhyloTree Build 17 [23] with an updated algorithm [24], and performs a contamination check and coverage analysis.

Recently, a novel approach to mtDNA NGS through whole genome sequencing (WGS) has been developed with the Applied Biosystems™ Precision ID mtDNA Whole Genome Panel (Thermo Fisher Scientific, USA) [25]. This library preparation kit, with 162 amplicons that target the whole mtDNA, is mostly used for forensic samples, achieving reliable results in often degraded samples [26,27,28,29] and with low DNA input (usually 0.1 ng of genomic DNA, but 6.25 pg [30] and, very recently, 0.6 pg [31] have been reported). Despite its most common use in forensic sciences, it has also been used in a range of other applications, from worldwide lineage studies [32], to rare mtDNA differences in monozygotic twins [33].

A disadvantage of this approach, however, is its use of a modified mtDNA reference with 16,649 bps, instead of the conventional 16,569 bps. Since the last amplicon corresponds to the last 28 bps and the first 80 bps in the widely used rCRS, the Precision ID mtDNA reference duplicates the first 80 bps of the rCRS at the end of the reference. Thus, most of its bioinformatic processing must be done opaquely in the enterprise software, Ion Torrent Suite™ Software (TSS), with the possible addition of other company-owned software [31,34,35,36]. In addition to producing an unorthodox reference [37], which impedes further processing in open-source bioinformatic tools, sensitivity analyses have set its heteroplasmy threshold level to 10% [30,38,39].

In this study, we aimed to establish an alternative fully customizable pipeline—PrecisionCallerPipeline (PCP)—for the Precision ID mtDNA Whole Genome Panel, which (**I**) Produced BAMs accurately mapped to the rCRS, allowing complete integration with existing open-source mtDNA pipelines; and (**II**) Revisited the TSS’s heteroplasmy threshold of 10%. Our approach yielded improved performance metrics in comparison with TSS, additionally enabling the detection of heteroplasmic variants at the 2.5% threshold.

On the basis of this extensive validation process, we compared our pipeline to TSS by investigating 50 clinical samples from an ongoing stroke cohort, where we further validated our approach.

## 2. Results

### 2.1. Pipeline Validation with Samples from the 1000 Genomes Project

We acquired three DNA samples previously sequenced within the 1000 Genomes Project [4] on Illumina HiSeq (Illumina, Inc., USA), with three different sequencing runs (more details in Section 4.1. Sample acquisition/collection): (**I**) Exome sequencing—exome (mean coverage: 420.59 reads/base pair); (**II**) Low-coverage sequencing—lowCov (mean coverage 3863.37 reads/base pair); and (**III**) High-coverage sequencing—highCov (mean coverage: 15079.28 reads/base pair). Samples were then processed and mixed at five different mixtures levels, ranging from 1 to 25% (Figure 1A), and sequenced on Ion Torrent™ Ion S5™ (Thermo Fisher Scientific, USA). The original sequenced samples and their mixtures then underwent bioinformatic analysis with our pipeline, the PrecisionCallerPipeline (PCP) (Figure 1B). For that, we aimed to optimize the protocol for the heteroplasmy threshold and the removal of nuclear insertions of mitochondrial DNA (NUMTs). Correspondingly, we ran the samples with three different NUMT removal approaches and 21 thresholds, ranging from 0.4% to 10% (Figure 1C), through the mutserve variant caller, an offline and command-line version of mtDNA-Server [17,40]. After data analysis and variant classification for primary and mixture samples (Appendix A), we determined the performance metrics (Appendix A) considering both Grade A variants, which were homoplasmic in the three independent sequencing runs, and Grade B variants, which were heteroplasmic in the same three sequencing runs (more details in Section 4.4. Data analysis). Our primary outcome was the highest possible F_1_ score.

NUMT removal immediately after the first alignment and before merging the BAM files (Figure 1B) was superior to the other NUMT removal approaches (Appendix A), while the 2.5% heteroplasmy threshold with NUMT removal before merging yielded the highest mean F_1_ score between the four different datasets—Primary Grade A, Mixture Grade A, Primary Grade B, and Mixture Grade B—(Appendix A).

Correspondingly, all BAM files were then run through the mutserve variant caller with a 2.5% threshold and NUMT removal before merging (Figure 1B). For simplicity, we denote samples by their simplified haplogroup identifier. Thus, HG00256 as H, HG01626 as T, and HG01757 as U. Similarly, names for mixtures follow an analogous pattern with the minor component being followed by its absolute mixture level and the major component. Therefore, *U0.01H* is sample HG01757 (U) at 1% mixed with sample HG00256 (H) at 99%.

Genome coverage and mappability, defined as the ability to read a base pair (bp) above a certain coverage threshold, were adequate in comparison to other sequencing runs in the primary analysis. Haplogroup assignment was also homogeneous for both PCP and the output from Ion Torrent Suite™ Software (TSS) (Appendix A and Figure 2).

Interestingly, PCP had an increased number of mutations in comparison with TSS (Figure 2B), despite a reduction in the mean coverage in PCP (Figure 2A). To better understand this phenomenon, we considered all samples analyzed on the Ion Torrent (Appendix A and Figure 3).

In PCP, coverage and mappability were uniform, haplogroup assignment was successful for mixtures up to 10%, and sample contamination was correctly detected, with 5% as the lowest detection limit (Appendix A and Appendix A). TSS correctly identified haplogroups up to 10% as well, albeit with errors in the contamination detection for unmixed samples U and T (lower than TSS’s own limit of ~10%), and a higher contamination detection limit of 10% (Appendix A).

In comparison with TSS, samples processed with PCP have a reduced total number of sequences (Appendix A and Figure 3A)—mean difference of 16.94%, 95% CI [15.98%–17.89%], an adjusted *p*-value of 1.43 × 10^−17^, and a significantly lower mean coverage (Appendix A and Figure 3B)—mean difference of 25.83%, 95% [24.97%–26.70%], an adjusted *p*-value of 3.87 × 10^−21^; paired t-tests adjusted with false discovery rate (FDR). This arises from our read trimming, NUMT removal, and quality control protocol, since, in comparison to the unaltered FASTQ files, the BAM files from TSS have the exact same number of reads and Phred score patterns (Figure 3D and Figure 3F).

In comparison with PCP, these results indicate that TSS achieves a higher coverage at the expense of a lack of read selection from the initial FASTQ files. Despite having significantly reduced coverage and fewer reads, more variants are called with PCP than with TSS (Appendix A and Figure 3C)—mean difference of 5.83, 95% CI [1.95–9.72], an adjusted *p*-value of 5.60 × 10^−3^; paired t-tests with FDR.

To further dissect this difference in the number of variants, we first looked at the variants in the primary analysis (Table A1, Appendix A, and Appendix A).

When comparing variants found with the Ion Torrent sequencing method to the ones previously found on the other three Illumina runs, we observe that: (**I**) PCP significantly increases the proportion of correctly found Grade A/B variants (true positives) in contrast to lost Grade A/B variants (false negatives), while no Grade A/B lost variant with PCP was picked up by TSS (Appendix A); (**II**) TSS significantly overestimates the variant level (VL) in found Grade A/B variants in comparison with PCP (Appendix A); (**III**) Despite PCP picking up more variants than TSS, the difference in the proportion of “novel” variants (false positives) was not statistically significant between the two pipelines (Appendix A).

We then looked at the variants in the mixture analysis (Figure 4, Table A2, Appendix A, Appendix A, Appendix A, Appendix A). Samples with a macro classification of “Mixed status” (more details in Appendix A) were excluded because of the impossibility of determining their origin.

Similar to the primary analysis, Grade A/B variants were best retrieved with PCP, in comparison with TSS, with a single variant found by TSS and lost with PCP (Appendix A). Moreover, true positive variants below 11% are completely absent in TSS, while PCP correctly picks up variants until its threshold of 2.5% (Figure 4 and Table A2).

Regarding the difference between the expected and the observed VL, we defined two different approaches:External comparison—the difference to the other Illumina sequencing methods (exome, lowCov, and highCov);Internal comparison—the difference within Ion Torrent, taking the primary sequencing as a reference.

Consequently, we also considered two different relative differences, where we divided each difference by its expected level in order to observe a percent variance per VL (Figure 4A, Appendix A). When performing a paired comparison regarding differences for Grade A/B variants (Appendix A), we observed no significant differences between TSS and PCP for raw VL differences taking found variants into consideration. However, when we considered lost variants, TSS performed worse in the external comparison (Appendix A). Regarding relative differences (Appendix A), we found no statistically significant differences between PCP and TSS.

In order to assess the similarity between the expected and the observed VLs (internal and external), we also performed linear models per mixture, considering Grade A/B variants for each pipeline, for each comparison (internal and external), and for each grade variant classification (A or B), as they might have varied between them. After extracting the adjusted R^2^ and the adjusted *p*-value with FDR, we created an indicator for correlation strength determined by the log10 of the ratio between the adjusted R2 and the adjusted *p*-value. Thus, a greater value would indicate a stronger correlation between both variables. PCP showed a significantly increased correlation indicator in both comparisons (internal and external) for Grade B variants, while no significant differences for Grade A variants were found (Figure 4B and Appendix A).

Since PCP appeared to have increased sensitivity in comparison with TSS, we then looked at the prevalence of novel variants (false positives). Interestingly, no differences were found (Appendix A), albeit with TSS showing novel variants at a higher VL, namely, novel variants already present in the primary analysis (Appendix A).

When we looked at the location of these novel variants in both analyses (primary and mixture), PCP had one mutation in 10/18 samples (4318T), one mutation in 8/18 samples (8649C), while the remaining four were isolated (2463G, 5752G, 6698G, 8152A). TSS had two mutations shared in 10/18 samples (310C, 10958C), one shared in 4/18 samples (14777C), one shared in 3/18 samples (8249C), and one shared in 2/18 samples (318C).

Looking into the different variant classifications (found, lost, novel, and primary novel variants), we noticed that: (**I**) PCP showed an increased rate (adjusted to the size of each region) of found variants throughout the genome, while maintaining a lower rate of lost variants (Appendix A and Appendix A); (**II**) Novel variants (primary and mixture) showed the lowest normalized coverage (normalized for the mean coverage of each sample in Appendix A) for both PCP and TSS, while the highest strand bias, calculated as a coverage ratio (forward vs. reverse), was observed in mixture novel variants for PCP, and in primary novel variants for TSS (Appendix A and Appendix A); (**III**) Since false variants might arise from NUMTs, we calculated the mean value of reported NUMTs based on two different databases [17,41] for each position—the highest rate of NUMTs was observed in primary novel variants for PCP, and in found variants for TSS (Appendix A and Appendix A); (**IV**) Regarding the proportion of variants in positions classified as a low-complexity region (LCR) [17], lost variants showed the highest proportion of LCR variants for PCP, while the same was true in primary novel variants for TSS (Appendix A and Appendix A); (**IV**) Finally, we tested the hypothesis that the distance to the “callable” extremity of each amplicon might be significantly different, depending on each variant class. The lowest distance was observed in primary and mixture novel variants for PCP, while the same was true for primary novel variants alone for TSS, although less consistently (Appendix A and Appendix A). Table 1 offers a summary for novel variants (false positives).

Ultimately, we performed a performance analysis for four different datasets: Primary Grade A, Mixture Grade A, Primary Grade B, and Mixture Grade B, with separate paired statistical tests for each dataset (Figure A1 and Appendix A). PCP achieved a higher sensitivity and F_1_ score in both the primary and mixture analyses for Grade B variants, without performing worse than TSS in the other indicators or in the other datasets (Appendix A).

In order to further validate our findings, we performed the same analyses with two other variant callers (freebayes [42] and varscan 2 [43]), which yielded overall overlapping results (Appendix A).

### 2.2. Pipeline Comparison with An Independent Set of 50 Clinical Samples

As a way of demonstrating the added value of our pipeline, we selected 50 independent samples from a prospective study in patients with ischemic stroke (more details in Section 4.1. Sample acquisition/collection) and analyzed them in parallel with TSS and PCP.

In PCP, genome coverage followed a similar pattern in all samples (Figure 5A), with an overall uniform mappability (Appendix A). The genome coverage was also in accordance with what is typical in Ion Torrent™ sequencing with the Precision ID mtDNA Whole Genome Panel [35]. The haplogroup was identical for both pipelines in 45/50 samples (Appendix A). In PCP, 3 samples were flagged for contamination in contrast to TSS, which had 11 samples, 10/11 were well below its ~10% threshold (Appendix A). In PCP, the number of variants varied extremely, from 12 to 503, while in TSS, they ranged from 10 to 80 (Appendix A). Thus, we eliminated Samples #21, #37, and #44 because of their contamination status, and Samples #2 and #3 because of their extremely high number of variants in PCP, from our subsequent analysis.

Consistent with what we described previously, while mean coverage was significantly decreased in PCP (Figure 5B)—mean difference of 30.87%, 95% CI [29.17%–32.58%], an adjusted *p*-value of 3.01 × 10^−34^, we still observed a higher number of variants with our pipeline (Figure 5D)—mean difference of 15.11%, 95% [10.07–20.16], an adjusted *p*-value of 2.97 × 10^−7^; paired t-tests adjusted with FDR.

This difference mostly derives from an increase in variants below TSS’s heteroplasmy threshold (Figure 5C): variants only present in PCP have a mean VL of 7.45%, while variants only present in TSS have a mean VL of 22.5%. Interestingly, PCP finds 98.31% of TSS’s variants, while TSS only finds 57.92% of PCP’s variants. Although not exclusively, when we consider discordant variants (variants not present in both pipelines) with more than one occurrence (Appendix A), we observe similar patterns as false positives in the previous analyses (Appendix A and Appendix A). In PCP, for example, we encounter the same variants seen in the previous analysis: 8649C (present in 25/45 samples) and 4318T (present in 20/45 samples). These two samples have a mean VL <5%, a low mean normalized coverage, and a small distance to the amplicon’s “callable” edge. Similarly, in TSS, we observe variants that were also previously found: 10958C (present in 17/45 samples) and 14777C (present in 2/45 samples). These two variants have a low mean normalized coverage, without other discernible features.

Regardless of the discordant variants, in the variants found in both pipelines, there is a statistically significant correlation between the VL with TSS, and with PCP (Appendix A)—adjusted R^2^ of 0.97 and a *p*-value < 2.2 × 10^−16^; linear regression model.

## 3. Discussion and Conclusions

In this study, we have developed a fully customizable pipeline—PCP—for the Precision ID mtDNA Whole Genome Panel, which produced BAMs accurately mapped to the rCRS and that has revisited the heteroplasmy threshold of the current gold standard method—TSS.

On the one hand, despite arising from the same set of samples and sequencing runs, the output from PCP and TSS was sufficiently different to yield significant changes between the two pipelines. In comparison with TSS, our method accomplished better performance metrics in previously sequenced samples, namely, higher sensitivity and F_1_ scores without a decrease in specificity and precision. This was mostly due to the additional detection of heteroplasmic variants with a VL above 2.5%, but below TSS’s ~10% threshold. Interestingly, this increase in the number of correct variants and the improved performance was achieved despite PCP having lower coverage and fewer sequences. Moreover, the same pattern of a higher number of variants at a lower threshold, notwithstanding a lower mean coverage and fewer sequences, was also observed in an independent set of clinical samples.

On the other hand, mtDNA differences between the same sample are not unheard of, as DNA polymerases, amplification protocols, sequencing runs, and variant callers are frequent sources of disparity in genomics [35,44]. Discerning sequencing errors/false positives from true heteroplasmic variants is a challenging task, usually achieved through post-sequencing curation, looking for signs of poor amplification, strand bias, mutations in LCR, and the presence of NUMTs [41,45,46,47,48,49].

Although sometimes ignored [50], heteroplasmic variants are ubiquitous [51,52,53,54] and show significant tissue-specificity [53,54]. Interestingly, low-level heteroplasmic variants (VL <10%) have shown matrilineal inheritance [55,56] with increased relevance in aging [57], and in clinical settings, particularly cancer [58,59]. Nonetheless, the reliability of these low-level variants has been a matter of debate [60,61,62] and, thus, current guidelines/workflows suggest a heteroplasmy threshold of 10%, based on the limitations of Sanger sequencing and electrophoresis technology [35,37,63,64,65,66].

False positives in both PCP and TSS showed a decreased normalized coverage, which indicates poor amplification in those regions, and were mostly transitions, which is in accordance with the literature [66,67]. However, the remaining putative mechanisms of error were very different between the two pipelines, and even showed differences within the same pipeline, depending on where the novel variants were found (unmixed vs. mixed samples). Correspondingly, in the analysis of the 1000 Genomes Project’s samples, no false positives were shared.

In parallel, discordant variants in the stroke cohort showed most of the variants previously flagged as false positives, albeit with many more variants only present in PCP at a low level, which had not been flagged before. This different mutational pattern from the samples sequenced within the 1000 Genomes Project might arise from the tissue-specificity of heteroplasmic variants, since blood, from where we extracted the DNA in our stroke cohort, has a lower level of heteroplasmy [53,54] and a very diverse mixture of mtDNA content [68], as it encompasses multiple cell subtypes.

Our pipeline offers multiple advantages in comparison with TSS. Firstly, it is in line with the Findable, Accessible, Interoperable, Reusable (FAIR) principles [69], increasingly important in the field of genomics [70]. Thus, our open-access approach makes it easier to implement validated and uniform bioinformatic protocols in genomics [3,71,72]. Secondly, it outputs a BAM file with the correct rCRS reference, thus allowing for integration into other NGS workflows and greatly facilitating variant annotation. This is particularly important when dealing with high-throughput datasets, where manual curation and inspection are not feasible nor efficient, thus favoring computational approaches to DNA variant analysis [73]. Finally, it is fully customizable. In a world of limited resources (personnel, time, samples, funding), our workflow is easily adaptable, depending on the focus of each research project and tissue mutational pattern. On the one hand, if one wishes to prioritize sample diversity in favor of replicates, variants might be called with PCP and filtered if not present in TSS, since PCP captures ~98% of all TSS variants with a very high VL correlation, and TSS discordant variants are likely to be false positives, in accordance with our analysis. On the other hand, if one prefers to sacrifice sample diversity in favor of VCM performance, the protocol for NUMT removal and heteroplasmy threshold optimization in unmixed and mixed samples yields an optimal workflow specific to that set of samples. Furthermore, the two approaches might be combined, or even expanded, into FASTQ quality control, alignment, or further variant annotation, with the addition of hypervariable segments [74], poly C-stretches [75], which are particularly difficult in Ion Torrrent™ sequencing [76], and the presence of homoplasmic or heteroplasmic variants at the HelixMTdb [67] (see Appendix A for an example). In summary, future clinical research will benefit from using our open-source bioinformatic processing since it keeps the advantages of the Precision ID library kit, particularly its low DNA input, while circumventing the limitations of TSS, namely, its modified reference sequence, proprietary nature, and 10% heteroplasmy threshold. For separating true variants from false variants, we also present a range of options that can be used in combination or not, depending on the samples and research focus: (**I**) A dedicated mixture optimization protocol; (**II**) Variant filtering based on normalized coverage, strand bias, the presence of NUMTs, among other parameters; (**III**) Filtering the output from PCP with TSS.

Nonetheless, our study also has a few limitations. Firstly, we used a limited set of samples for the primary and mixture analysis, which are not representative of the range of possible haplogroup combinations. Secondly, samples from our stroke cohort were not sequenced in duplicate nor mixed, which did not allow for a replication of the performance metrics we did previously. Finally, we did not consider insertions nor deletions and, thus, we are unable to provide any recommendations for the analysis of those variants with our pipeline.

Overall, we have developed the first open-source alternative to the enterprise software, Ion Torrent Suite™ Software (TSS), for the Precision ID mtDNA Whole Genome Panel, with improved performance metrics, and with an output in the correct rCRS reference. Since the majority of existing mtDNA bioinformatic tools [6,7,8,9,10,11,12,13,14,15,16,17,18,19,20,21] are only compatible with the correct rCRS format, and the few tools that perform variant calling in samples sequenced with the Precision ID library kit [31,34,35,36] keep TSS’s modified reference sequence, PCP is currently the sole option that bridges this bioinformatic gap, allowing for the universal variant calling of samples sequenced with the aforementioned library.

The herein presented pipeline, PCP, is available in the GitHub repository, https://github.com/filcfig/PCP (accessed on 1 November 2021). Additionally, we provide the generated data for validation (see Data Availability Statement).

## 4. Materials and Methods

### 4.1. Sample Acquisition/Collection

In order to validate our pipeline—PrecisionCallerPipeline *(PCP)*, we acquired three cell-line samples from the Coriell Institute for Medical Research (Camden, NJ, USA), i.e., HG00256, HG01626, and HG01757. These samples have been sequenced within the 1000 Genomes Project [4] on Illumina HiSeq (Illumina, Inc., San Diego, CA, USA), with three different sequencing runs, mostly targeted at nuclear DNA: (**I**) Exome sequencing—exome; (**II**) Low-coverage sequencing—lowCov; and (**III**) High coverage sequencing—highCov.

With the aim of evaluating the sensitivity, specificity, precision, and F_1_ score (harmonic mean of precision and sensitivity) [17,44] with different variant detection thresholds, we performed three different mixtures at five different mixture levels (1, 2, 5, 10, and 25%):Mixture HT—Minor component: HG00256 (haplogroup H5b2, short identifier H) + Major component: HG01626 (haplogroup T2a1b1a1b, short identifier T);Mixture TU—Minor component: HG01626 (haplogroup T2a1b1a1b, short identifier T) + Major component: HG01757 (haplogroup U4a, short identifier U);Mixture UH—Minor component: HG01757 (haplogroup U4a, short identifier U) + Major component: HG00256 (haplogroup H5b2, short identifier H).

DNA concentration measured prior to shipping ranged from 307 to 331 ng/μL. Thus, DNA mixtures were volume-based (Figure 1A).

As an independent set of samples, we analyzed 50 samples derived from a prospective stroke cohort of patients at the Hospital de Santa Maria, Centro Hospitalar Universitário Lisboa Norte. Blood samples were collected within 72 h of hospital admission and all cases were reviewed and confirmed by trained neurologists. Inclusion criteria were: (**I**) Ischemic stroke; (**II**) Age ≥ 18 years old; (**III**) Blood samples collected up to 72h after symptom onset. The exclusion criteria were: (**I**) Active cancer diagnosis; (**II**) Previous cerebral revascularization surgeries; (**III**) Modified Rankin score [77] ≥ 5. The approval of the institutional review board (IRB) was conceded by the Comissão de Ética do Centro Académico de Medicina de Lisboa (reference 435/16, approved on 14 December 2016), informed consents were given by every subject, and the study followed the standards of the Declaration of Helsinki. DNA extraction was performed after PBMC isolation with the QIAamp^®^ DNA Blood Midi Kit (QIAGEN GmbH, Hilden, Germany), according to the manufacturer’s instructions.

### 4.2. DNA Sequencing

Samples were sequenced without prior long-range PCR (LR-PCR) with the Applied Biosystems™ Precision ID mtDNA Whole Genome Panel (Thermo Fisher Scientific, Waltham, MA, USA), in conjunction with the Ion Torrent™ Ion S5™ (Thermo Fisher Scientific, Waltham, MA, USA), in accordance with the manufacturer’s instructions. Briefly, DNA was quantified with a Qubit^®^ 3.0 fluorometer (Thermo Fisher Scientific, Waltham, MA, USA) and samples were diluted to 0.0067 ng/μL for an input of 0.1 ng of genomic DNA in 15 μL. Libraries were prepared using the Ion Chef™ automated protocol, and samples were then run on 530™ chips with the Ion Torrent™ Ion S5™ at Ipatimup—Instituto de Patologia e Imunologia Molecular da Universidade do Porto (Porto, Portugal).

### 4.3. Bioinformatic Processing

#### 4.3.1. PrecisionCallerPipeline (PCP)

Our PCP pipeline automatically takes the FASTQ files from the sequencing facility and outputs fully aligned BAM files mapped to the commonly used reference sequence, rCRS [22]. We use a workflow based on Snakemake [78] that uses: (**I**) Awk, for SAM file editing [79]; (**II**) BEDTools, for BAM to FASTQ conversion [80]; (**III**) BWA-MEM, for read alignment [81]; (**IV**) Pycision, for amplicon delimitation and selection [34]; (**V**) SAMtools for BAM conversion, sorting, indexing, and merging [82]; and (**VI**) Trimmomatic for read quality control and trimming [83] (Figure 1B). Removal of NUMTs was tested before and after final BAM merging with RtN! [47] (Figure 1C).

Samples were processed in a Linux-based system. For simplicity, a predetermined file structure and the necessary files (except for the files from external software) may be downloaded from https://github.com/filcfig/PCP.git (accessed on 1 November 2021). A separate Snakefile (Snakefile_no RtN) is also provided to run the samples without the removal of NUMTs.

Read quality analysis was performed with FastQC [84] and MultiQC [85]. For read quality control, we opted to crop reads at 160 base pairs (bps), taking into account: (**I**) The visual inspection of Phred score patterns per read bp (Figure 3D); (**II**) The length range of 73–137 bps for the known “callable” sections of each amplicon [34], also considering the company-reported amplicon average length of 163 bps, as the exact coordinates of the amplicon themselves are unknown.

Variant calling was performed with freebayes v1.3.5 [42], mutserve v2, the command line interface and successor of the mtDNA-Server pipeline [17,40], and VarScan 2 v2.3.7 [43]. Initially, we used 21 different heteroplasmy thresholds, ranging from 0.4% to 10.0% (Figure 1C and Appendix A). After optimization, the maximum F_1_ score was achieved with a heteroplasmy threshold of 2.5% and with NUMT removal before the final BAM merge (Figure 1B, Appendix A); this processing was then used in all variant calling methods (VCMs). For the Illumina sequencing runs (exome, lowCov, and highCov), a threshold of 0.4% was maintained as a reference. We used very similar parameters as [44], with the exception of a base quality score of 20 for all VCMs, as well as, for freebayes, where we used *“--ploidy 1 --pooled-continuous”*. Similar to [44], variants in positions 302–315 (position 310 was blacklisted in our analysis), 523–524, and 3104–3110 were excluded. Only single nucleotide substitutions were considered, and variants below the established threshold were filtered.

Haplogroup calling was carried out through HaploGrep v2.4.0 [18], and a contamination check was done with Haplocheck v1.3.3 [40], based on the output from mutserve.

#### 4.3.2. Ion Torrent Suite™ Software (TSS)

For the current gold standard, data from each run was processed using the Ion Torrent™ specific pipeline software, Ion Torrent Suite™ Software (TSS), using the reference sequence PrecisionID_mtDNA_rCRS, and target regions PrecisionID_mtDNA_WG_targets with the plugins CoverageAnalysis and VariantCaller. FASTQ and BAM files were generated using the plugin FileExporter. The software versions ranged from v5.8, v5.10, and v5.12, according to the date of each run.

We compiled the VCF files arising from the sequencing runs and corrected all positions > 16,569 to the first 80 bps in the rCRS. When we observed equal variants with different coverages and variant levels (VLs), particularly in the first 80 bps, we calculated the mean coverage and VL per mutation. Only single nucleotide substitutions were considered.

After exporting the corrected variants in a VCF format, where GT 1/0 was defined for VL ≥ 90%, we ran the samples through HaploGrep and Haplocheck, similar to PCP.

### 4.4. Data Analysis

Data analysis was performed with R version 4.1.1 [86] in RStudio [87] with the packages extrafont [88], infer [89], magick [90], patchwork [91], readxl [92], remotes [93], scales [94], svglite [95], and tidyverse [96], as well as Excel 2016 (Microsoft Corporation, Redmond, WA, USA).

Analyses were performed in parallel for PCP’s output and TSS’s output. For the primary analysis, variants from the resequenced unmixed samples were compared to the ones identified in Illumina for the exome, lowCov, and highCov sequencing runs, and classified according to their reliability:Grade A variants: homoplasmic variants (mean variant level ≥ 95%) found in both highCov and lowCov, regardless of exome;Grade B variants: heteroplasmic variants (mean variant level ≥ 0.4% and ≤ 95%) found in both highCov and lowCov, regardless of exome, or found in highCov plus exome, or lowCov plus exome;Grade C variants: found in a single sequencing run;Novel variants: found in the Ion Torrent runs only.

For the mixture analysis, variants from the 15 mixtures were classified according to the interaction between both VL and primary variant classifications (explained previously) in both components (minor and major). We began with a manual curation of all 512 theoretical combinations, which gave rise to 128 possible scenarios, grouped in 47 micro classes, 7 meso classes, and 5 macro classes (Appendix A). In this case, due to the possibility of shared variants—variants in the same position in two different samples—having two different classifications, we assumed that lower grade variants would prevail in combinations. Hence, a shared variant with both Grade A and Grade C variant classifications would receive a Grade C classification.

## Figures and Tables

**Figure 1 ijms-22-12031-f001:**
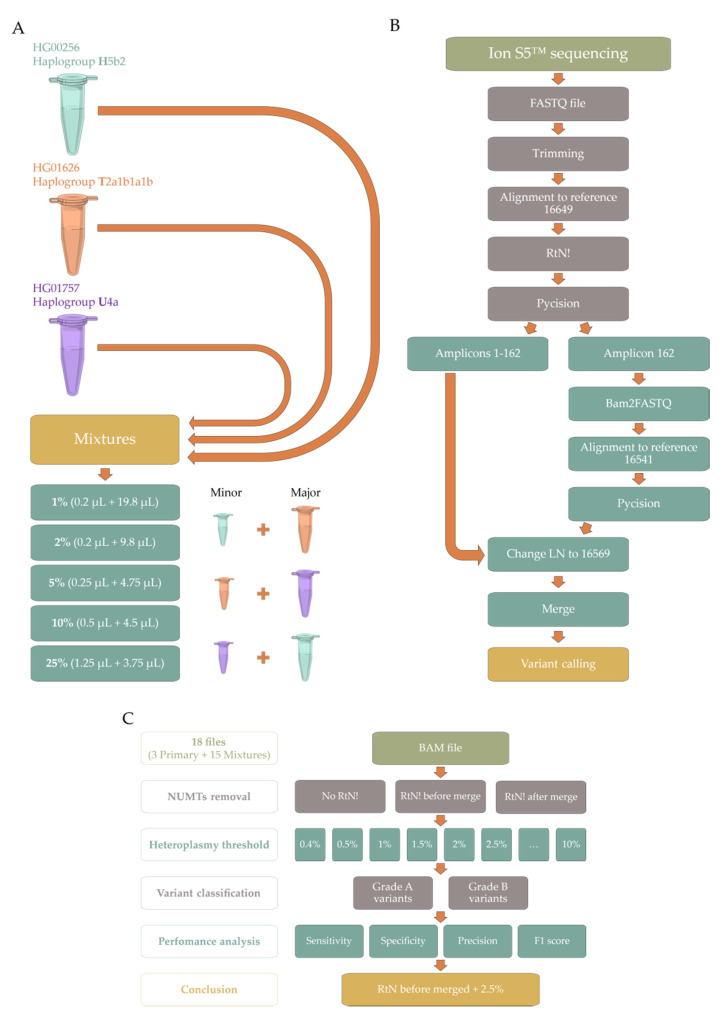
Schematic representation of our workflow. (**A**) Wet lab approach to performing the 15 mixtures from the Coriell Institute for Medical Research’s samples, which have been analyzed within the 1000 Genomes Project—templates from Servier Medical Art (CC BY 3.0) were adapted, and are freely available at https://smart.servier.com (accessed on 1 November 2021); (**B**) Visual representation of the PrecisionCallerPipeline (PCP); (**C**) Visual representation of the optimization approach to NUMT removal and heteroplasmy threshold. Abbreviations: NUMTs—nuclear insertions of mitochondrial DNA.

**Figure 2 ijms-22-12031-f002:**
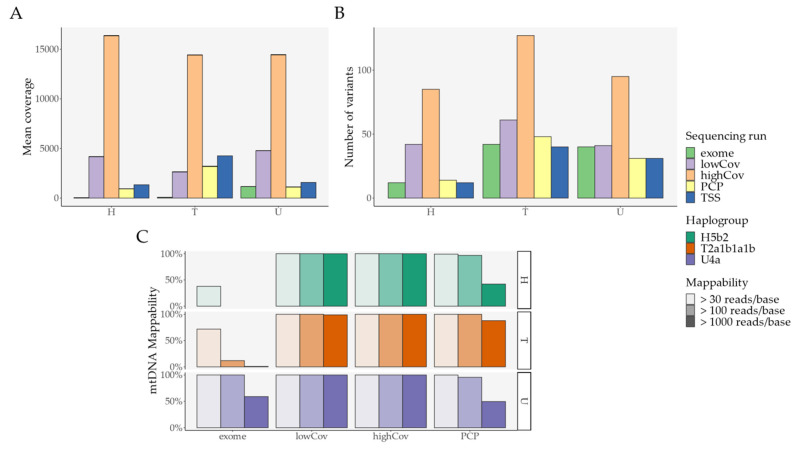
Primary analysis: Overall look; variant threshold was set at 0.4% for exome, lowCov, and highCov, and 2.5% for PCP. (**A**) Mean coverage per sample and per sequencing run/pipeline; (**B**) Number of variants per sequencing run/pipeline; (**C**) Mappability and haplogroup assignment per sequencing run/pipeline (TSS is not shown since it does not output raw data). Error bars denote standard error of the mean (SEM). Abbreviations: PCP—PrecisionCallerPipeline; TSS—Ion Torrent Suite™ Software.

**Figure 3 ijms-22-12031-f003:**
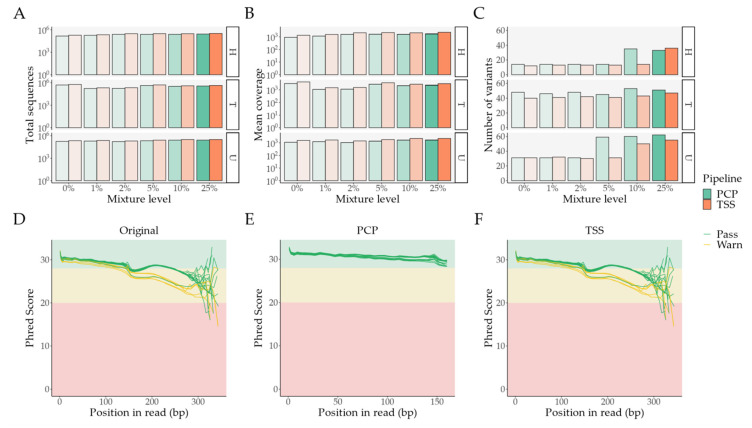
Ion Torrent sequencing: Overall look at differences between PCP and TSS; a 0% mixture level denotes unmixed samples. (**A**) Total number of sequences per pipeline and major sample contributor; (**B**) Mean coverage per pipeline and major sample contributor; (**C**) Number of variants per pipeline and major sample contributor; (**D–F**) MultiQC’s Phred score per base pair in the unprocessed FASTQ files, PCP’s BAM to FASTQ files, and TSS’s BAM to FASTQ files, respectively. Abbreviations: bp—base pair; PCP—PrecisionCallerPipeline; TSS—Ion Torrent Suite™ Software.

**Figure 4 ijms-22-12031-f004:**
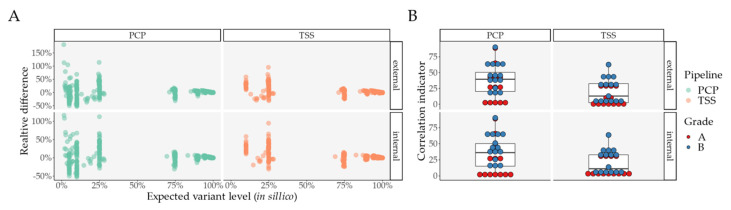
Mixture analysis: similarity in Grade A/B variants between expected and observed variant levels, for internal and external comparisons, per pipeline. (**A**) Relative difference between expected and observed variant level, per pipeline, and per internal and external comparison; (**B**) Correlation indicator for internal and external comparisons, per pipeline, and per variant grade. Abbreviations: PCP—PrecisionCallerPipeline; TSS—Ion Torrent Suite™ Software.

**Figure 5 ijms-22-12031-f005:**
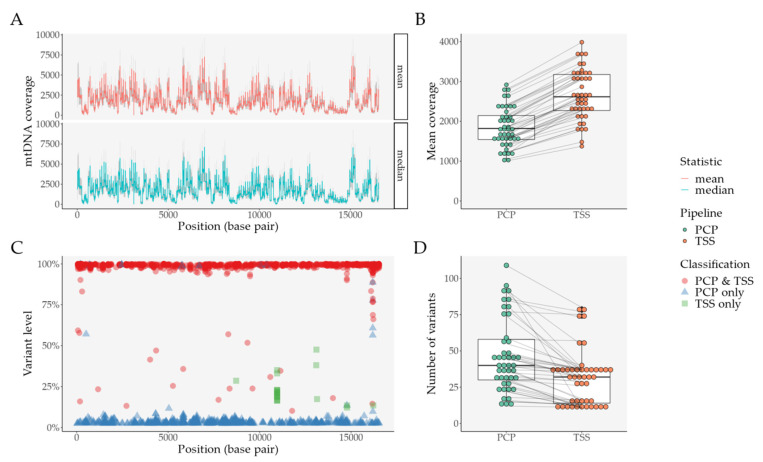
Comparison between PCP and TSS in an independent set of samples (after filtering, *N* = 45) derived from clinical practice (an ischemic stroke prospective cohort). (**A**) Coverage per base pair with PCP data, per statistic (mean and median)—the grey area represents the standard deviation of each statistic per base pair; (**B**) Mean coverage per sample and per pipeline—lines connect the same sample; (**C**) Distribution of variants per base pair and per variant classification—for variants found in PCP and TSS, mean variant level was plotted; (**D**) Mean number of variants per sample and per pipeline—lines connect the same sample. Abbreviations: PCP—PrecisionCallerPipeline; TSS—Ion Torrent Suite™ Software.

**Table 1 ijms-22-12031-t001:** Novel variants (false positives): Summary characteristics in the mutserve variant calling method (VCM).

Variable	PCP		TSS	
	Primary Novel Variants	Mixture Novel Variants	Primary Novel Variants	Mixture Novel Variants
Normalized coverage	↓	↓↓	↓↓↓	↓↓
Coverage ratio	-	↑↑↑	↑↑↑	↑
Number of NUMTs	↑↑↑	↓	-	↓↓
LCR prevalence	↓↓↓	↓↓↓	↑↑↑	-
Distance to amplicon edge	↓↓↓	↓↓	↓	-

Using found variants (true positives) as a reference: “-“ denotes nonsignificant changes; “↑” or “↓” denote significant changes in the 25–50% range; “↑↑” or “↓↓” denote significant changes in the 51–75% range; and “↑↑↑” or “↓↓↓” denote significant changes > 75% (more details in Appendix A). Abbreviations: PCP—PrecisionCallerPipeline; TSS—Ion Torrent Suite™ Software; NUMTs—nuclear insertions of mitochondrial DNA; LCR—low-complexity region.

## Data Availability

The data presented in this study, namely, the data generated with the samples previously sequenced within the 1000 Genomes Project, are openly available in Zenodo [97] at doi:10.5281/zenodo.5524539. Data analyzed within the stroke cohort are available upon request from the corresponding author; the data are not publicly available because of ethical and privacy restrictions.

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
