# Peer review of "From Forensics to Clinical Research: Expanding the Variant Calling Pipeline for the Precision ID mtDNA Whole Genome Panel"

_ijms, 2021, doi:10.3390/ijms222112031_

Round 1

Reviewer 1 Report

In the manuscript "From forensics to clinical research: expanding the variant calling pipeline for the Precision ID mtDNA Whole Genome Panel" Cortes-Figueiredo and colleagues describe a new pipeline for processing samples sequenced on Ion Torrent sequencing systems with the Precision ID library kit. In particular, the authors suggest an open-source alternative to the popular proprietary Torrent Suite™ Software (TSS).
The study design is rigorous and well conducted however, the convoluted structure of the paper makes it difficult to read and follow the articulated results. Although I believe that the procedure is valid, I would like the authors to discuss more in detail the advantages and disadvantages of the PrecisionCalllerPipeline (PCP) method compared to the existing not-proprietary bioinformatic tools for mtDNA analysis mentioned in the paper.

Reviewer 2 Report

In the manuscript, Cortes-Figueiredo et al present their work on a novel pipeline, PCP, to call variants on mtDNA genome. They compare the method with the previous used, and found their method is more powerful. The results are well presented and the conclusion was supported by the data. My concerns is the potential issue of false positive: in the application on 50 clinical samples, the new method detected twice number of variants as its counterpart method, and from the description, I couldn't find a clear message whether the new detected variants are true or false positive. The authors should give some more explanation, and give the guidance in result interpretation considering its future application in real clinical trial. 
